# Priorities in the Prevention Strategies for Medication Error Using the Analytical Hierarchy Process Method

**DOI:** 10.3390/healthcare10030512

**Published:** 2022-03-11

**Authors:** Siin Kim, Hyungtae Kim, Hae Sun Suh

**Affiliations:** 1College of Pharmacy, Kyung Hee University, Seoul 02447, Korea; siin@khu.ac.kr; 2College of Pharmacy, Pusan National University, Busan 46241, Korea; jackja@pusan.ac.kr

**Keywords:** analytic hierarchy process, medication error, patient safety, preventive strategies, prioritization

## Abstract

As medication error is inherently “preventable”, we should try to minimize errors to improve patient safety and quality of care. The aim of this study was to prioritize strategies to prevent medication errors using the analytic hierarchy process (AHP) method. The hierarchy structure consisted of three stages: goal of the decision, decision criteria, and alternatives. Ten experts of patient safety research or clinical pharmacology compared each pair of criteria and alternatives and assigned a nine-point numerical scale. We used the eigenvector method to aggregate the pairwise comparisons obtained from experts and to estimate the weights of each criterion and alternative. Among the decision criteria, system improvement in reporting was the most preferred criterion, followed by cultural improvement and system improvement in the counterplan. The preferred alternative was a counterplan by healthcare institutions, followed by a change from a blame culture to safety culture and the building of a reporting system. A sensitivity analysis indicated that priorities were generally robust in the methods used for calculating the integrated matrices. We have suggested the priority of preventive strategies against medication errors using the AHP method. The prioritization of preventive strategies could help policymakers understand current needs and therefore develop evidence-based policies on patient safety.

## 1. Introduction

According to the definition of the National Coordination Council for Medication Error Reporting and Prevention, a medication error is defined as “a preventable event that may cause or lead to inappropriate medication use or patient harm while the medication is in the control of the health care professional, patient, or consumer” [1]. Some medication errors result in adverse drug events (ADEs), called “preventable ADEs”. Some medication errors do not result in adverse events but still have the potential to cause adverse events; these are called “potential ADEs” [2].

The Institute of Medicine reported that more than 1.5 million preventable ADEs occurred annually in the United States [3]. In a previous study, there were 7.3 preventable events (including both preventable ADEs and potential ADEs) per 100 hospital admissions, and the incidence of potential ADEs was three times that of preventable ADEs [4]. In Korea, of all drug-related damages reported to the Korea Consumer Agency increased between 2010 and 2014, 9.2% of them resulted from medication error [5]. Medication errors result in a considerable economic burden, which was estimated to be GBP 98.5 million per year in England [6].

Because medication errors are “preventable”, the errors can be minimized by effort, leading to improvements in safety and quality of care [3,7]. However, medication errors cannot be prevented merely by an individual’s effort. In fact, a majority of medication errors result from fundamental causes such as human factors, defects in the system, and inadequate healthcare products [8]. Owing to their diverse causes, there are numerous strategies to prevent medication errors. The Food and Drug Administration (FDA) of the United States currently operates the FDA Adverse Event Reporting System (FAERS), a database that collects drug adverse events and medication errors. The data accumulated in FAERS are analyzed and evaluated by the Center for Drug Evaluation and Research and Center for Biologics Evaluation and Research (CBER) for postmarketing surveillance [9]. The National Health Service of the United Kingdom implemented national campaigns, such as ‘Patient Safety First’ and ‘Sign up to Safety’, and encouraged incident reporting by assuring anonymity and giving an incentive [10]. Moreover, various education programs have been delivered to patients vulnerable to medication errors, including elderly, asthma, and diabetes patients, and have been demonstrated to be effective for preventing errors [11,12,13].

Decision makers of patient safety policies have difficulty in setting priorities between various alternatives, considering the relevant evidence and necessity of each alternative, and decision-making becomes more complicated when involving several individuals that have different priorities for the alternatives [14]. To make a decision systematically and transparently, the analytic hierarchy process (AHP) can be utilized. The AHP, first developed by Saaty [15], is a multi-criteria decision analysis technique [16]. AHP requires a hierarchical structure of the criteria and alternatives related to the goal. The advantages of AHP are that it uses a ratio scale to compare the pairwise of items at the same level, and it enables the measurement of the consistency of judgement [17]. Considering that decision-making in healthcare inevitably involves expert judgements, multi-criteria decision analysis techniques such as AHP can enhance the transparency of the decision-making process in a systematic and explicit way [16]. Therefore, this study aimed to prioritize strategies to prevent medication errors in South Korea using the AHP method.

## 2. Materials and Methods

To prioritize strategies to prevent or reduce medication errors, we used an AHP method that consisted of three steps: decomposition of the structure, comparison of judgements, and hierarchical composition of priorities [15]. This study was approved by the Institutional Review Board of Pusan National University (PNU IRB/2020_58_HR).

### 2.1. Decomposition of the Structure

In the hierarchical structuring step, we first investigated previous literature and the systems in various countries (i.e., Australia, Canada, New Zealand, United Kingdom, United States of America) to prevent medication errors [18,19,20,21,22,23,24,25,26,27,28,29,30,31,32,33,34,35,36,37,38,39,40]. Three independent reviewers categorized the elements in a hierarchical manner and reached a consensus by discussion. The fourth reviewer independently assessed the structure to ensure the validity of the structure. The hierarchy structure consists of three stages. The first stage of the structure was the goal of the decision, and the second stage was the decision criteria. Finally, the third stage consisted of alternatives within each criterion [41]. The concept of medication error used in the hierarchy structure was based on previous studies on the definitions and types of medication errors in various countries [42,43].

### 2.2. Comparison of Judgements

Ten experts in South Korea, selected as respondents, compared each pair of criteria and alternatives. Because both clinical and institutional perspectives were needed to set effective strategies for preventing medication errors, we recruited five experts who had experience in patient safety research and medication errors, and five experts who had experience in clinical pharmacotherapy. The characteristics of the experts are summarized in Table 1. Each respondent assigned a nine-point numerical scale (1 = equal importance, 3 = moderate importance of one over another, 5 = essential or strong importance, 7 = very strong importance, 9 = extreme importance, and 2, 4, 6, and 8 for intermediate values between the two adjacent judgements) to compare each pair of criteria and alternatives [44,45]. Each respondent evaluated each pair of criteria first and then made a comparative assessment among alternatives within each criterion [46].

### 2.3. Hierarchical Composition of Priorities

Among the various methods for estimating the weights from the results of the pairwise comparison matrix, the eigenvector method was used because this was the only way to estimate the consistency of the judgement of respondents in a pairwise comparison [44]. We estimated the normalized weights of each criterion and alternative using geometric mean calculated in every row of the pairwise comparison matrix. To examine the consistency of each expert’s judgement, the consistency ratio (CR) was calculated by dividing the consistency index (CI) by the random index (RI). The CI represents the average of the remaining eigenvalues from dividing λ_max_ (the maximum eigenvalues of the pairwise comparison matrix) by the size of the pairwise comparison matrix minus one. The RI was defined according to the size of the pairwise comparison matrix (*n*) (i.e., RI = 0 for *n* = 1, 2; RI = 0.58 for *n* = 3; RI = 0.90 for *n* = 4; RI = 0.58 for *n* = 5) [47,48]. A CR less than 0.2 was regarded as the permissible level of consistency in experts’ judgements [49,50]. To aggregate each pairwise comparison from the different experts, we applied three approaches [51]. Firstly, we aggregated all the matrices of the pairwise comparisons by geometric mean and then calculated the comprehensive weight using the eigenvalue method [52]. This method, which is the most widely accepted in the AHP method, was used for base-case analysis. Secondly, we calculated the weight of each individual pairwise comparison by applying the eigenvalue method and then aggregated the weights using the geometric mean. Finally, we calculated the weight of each individual pairwise comparison using the eigenvalue method and then aggregated the weights using the arithmetic mean. The latter two methods were used for sensitivity analyses. The approach in the base-case analysis is primarily used when information or previous research on the topic is rare, and the other approaches are used to place an emphasis on the expertise of individual responders involved in decision making [53]. The Pearson correlation coefficient was calculated to assess the concordance between ranks derived from the base-case analysis and the ranks derived from the sensitivity analysis.

We calculated all the weights of alternatives by multiplying each weight of the criterion and the weights of alternative under the criterion. A criterion or alternative with a higher weight was regarded as having a higher priority. The data were analyzed using Microsoft Office Excel software (version 2010, Microsoft Corporation, Redmond, WA, USA).

## 3. Results

The final version of the hierarchy structure consisted of five criteria: cultural improvement, system improvement in reporting, system improvement in cause analysis, system improvement in counterplan, and system improvement in assessment (Figure 1). Two to five alternatives were suggested as possible strategies to meet the corresponding criteria. The definitions of each criterion and alternative are presented in Table 2.

Pairwise comparisons of criteria and alternatives were conducted by ten healthcare professionals who were deeply experienced in healthcare policy or clinical service. All comparisons between criteria (Table 3) or alternatives (Appendix A) were incorporated into the integrated matrices. Some comparisons with CR larger than 0.2 were excluded from the analyses.

Normalized weights of criteria and alternatives were calculated from the integrated matrices and are shown in Table 3 and Table 4, respectively. The best preferred criterion was system improvement in reporting, and the best preferred alternative was counterplan by healthcare institutions (Table 5).

We conducted sensitivity analyses to check the robustness of the priorities calculated from the AHP using various methods for calculating the integrated matrices. As mentioned in the methods section, there are two other methods to calculate weights: aggregating individual matrices using the geometric mean and aggregating individual matrices using the arithmetic mean. Of the two methods, the former (Pearson correlation coefficient = 0.974) resulted in priorities that corresponded more with the base-case analysis than the latter (Pearson correlation coefficient = 0.819) (Table 5).

## 4. Discussion

In this study, we explored preventive strategies against medication errors and prioritized them using the AHP method. Among the decision criteria, system improvement in reporting, cultural improvement, and system improvement in counterplan comprise the main priorities with similar weights, whereas the system improvement in cause analyses and assessment showed lower weights. This might suggest that experts in patient safety and clinical pharmacotherapy feel the necessity of action plans rather than evaluation plans. Among the alternatives, counterplan by healthcare institutions, changing from a blame culture to a safety culture, and building of reporting systems had high weights. The results of the sensitivity analyses indicated that the priorities of decision criteria and alternatives were generally robust in aggregating approaches.

To our knowledge, this is the first study to provide evidence for priority setting in prevention strategies against medication errors using the AHP method. Although several studies have investigated strategies for the prevention of medication errors, most of them discussed the strategies in a descriptive way based on the literature review, and a few of them evaluated the priority of strategies using survey data. Matti et al. conducted a survey of staff at neonatal intensive care units in Australia and New Zealand to understand the prevention strategies utilized in clinical practice [54]. The survey found that smart infusion pumps, ward-based pharmacists, and the administration calculation test for nursing staff were the most frequently used strategies to prevent medication errors. Fortescue et al. examined the types of medication errors and strategies that might have prevented the potentially harmful errors in 1020 pediatric inpatients [55]. The study reported that the most effective interventions to prevent potentially harmful errors were ward-based pharmacists, improving communication between healthcare providers, and computerized physician order entry combined with clinical decision support system. Contrary to previous studies, our study assessed the priority of prevention strategies at a national level rather than an institutional level. However, our study demonstrated that counterplan by healthcare institutions was the most important alternative, highlighting the need for developing evidence-based strategies for healthcare institutions.

AHP is one of the most popular methods for solving complex multi-criteria decision-making problems because of its intuitiveness and ability to analyze both qualitative and quantitative criteria [56]. Therefore, many studies have utilized AHP to suggest evidence for decision making in healthcare problems. Hsieh et al. studied the error factors in the emergency department using a fuzzy technique for order preference by similarity to an ideal solution (TOPSIS) and AHP [57]. Fuzzy TOPSIS was utilized to rank the important error factors, and AHP was utilized to derive the criteria weights in TOPSIS. This study suggested the most important error factors, such as decision errors and crew resource management. Singh et al. explored the optimal management strategy in patients with sore throat using AHP [58]. The results indicated that the priorities among the five management strategies (i.e., no test and no treatment, rapid strep, culture, rapid strep and culture, and empiric treatment) depend on the likelihood of group A streptococcal infection and clinical judgments on the relative importance of the decision criteria. Reddy et al. introduced AHP to prioritize public health topics for guidance development [16]. The pilot test suggested that AHP could be a useful approach for prioritizing topics for guidance by structuring the decision process in a transparent way.

Decision-making on health policy often involves qualitative assessment that is inherently subject to ambiguity and implicit judgement. Multiple criteria decision analysis (MCDA) helps decision makers set priorities between the alternatives and make optimal decisions by providing a structured and explicit approach [14]. Therefore, MCDA has been applied to decision-making regarding healthcare problems at various levels, such as national, international, regional, or hospital level [59]. Considering that healthcare problems have a large impact on society, from quality of life in individuals to socioeconomic burden related to a certain disease, decisions should be made in a reasonable way. In that sense, MCDA such as AHP can be a useful tool that can help policy makers in the healthcare sector pursue evidence-based policymaking. Therefore, the number of studies using MCDA in the healthcare research field has steadily increased between 1990 and 2017; AHP (41%) was the most widely used method among the MCDA methods, followed by fuzzy logic (21%) and EVIDEM (12%) [60]. In this study, we used AHP because it enables structured decision making based on intuitive judgement of experts, reflecting their expertise and experience with patient safety. In addition, AHP has an advantage in its simplicity of application compared to other methodologies that use mathematical models. It uses a relative scale (i.e., standardized measurement unit) for pairwise comparisons, enabling efficient data accumulation and further analyses in case of conducting successive research. However, as the AHP uses linguistic expression to measure the responder’s decision, it inevitably involves linguistic uncertainty [61]. To overcome this uncertainty, the fuzzy AHP that combines the AHP and fuzzy set theory can be utilized in future research.

This study has several limitations. First, the priority of the strategies to prevent medication errors could change if the survey was conducted with a large number of experts. Further research is needed to confirm the findings of this study by involving various experts in the field of patient safety. Second, although we tried to develop a hierarchical structure with mutually exclusive items, some items might be dependent on each other. Third, when pairwise comparisons between criteria or alternatives were incorporated into the integrated matrices, some comparisons had a CR larger than 0.2. When considering this, it suggests a possible inconsistency in the experts’ judgement. We excluded the comparisons with CR larger than 0.2 from the integrated matrices.

## 5. Conclusions

We have suggested the priority of preventive strategies against medication errors using the AHP method, which enables structured decision-making based on expertise and experience of patient safety experts. Among the strategies, system improvement in reporting and counterplan by healthcare institutions are likely to play an important role in preventing medication errors. This priority could help policymakers understand the current needs and develop evidence-based policies on patient safety.

## Figures and Tables

**Figure 1 healthcare-10-00512-f001:**
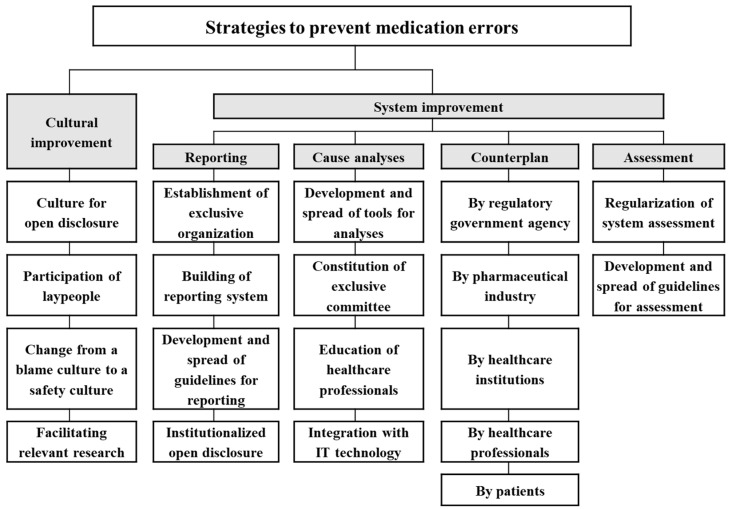
The hierarchy structure of strategies to prevent or reduce medication errors.

**Table 1 healthcare-10-00512-t001:** Characteristics of experts who compared each pair of criteria and alternatives regarding prevention strategies on medication errors.

Characteristics	Experts on Patient Safety Research (*n* = 5)	Experts on Clinical Pharmacotherapy (*n* = 5)
Female	3 (60%)	5 (100%)
Age (mean)	47.8 years	39.4 years
Specialty		
Medicine	2 (40%)	0 (0%)
Pharmacy	3 (60%)	5 (100%)
Affiliation		
Academy	3 (60%)	0 (0%)
Public institution	1 (20%)	0 (0%)
Medical institution	1 (20%)	5 (100%)
Work experience (mean)	16.6 years	13.2 years
Academic degrees		
Bachelor’s degree	0 (0%)	2 (40%)
Masters degree	0 (0%)	1 (20%)
Doctoral degree	5 (100%)	2 (40%)

**Table 2 healthcare-10-00512-t002:** Definitions of criteria and alternatives.

Criteria	Alternatives	Definition
Cultural improvement	Culture for open disclosure	To establish a culture that enables open disclosure between healthcare professionals and patients
	Participation of laypeople	To induce laypeople to participate actively in safe use by providing safe use information and running campaigns for spontaneous reporting of medication errors
	Change from a blame culture to a safety culture	To regard medication errors as a systemic problem and work together to find solutions instead of blaming an individual healthcare professional
	Facilitating relevant research	To encourage research on safety culture or safety policy
System improvement in reporting	Establishment of exclusive organization	To establish an exclusive organization that manages medication error reporting and assesses the current status of medication error regularly, thus leading to system improvement
	Building of reporting system	To establish a structured, national reporting system for patient safety event that encompasses adverse event and medication error
	Development and spread of guidelines for reporting	To develop and disseminate standardized guidelines for medication error reporting for specific population (e.g., the public, healthcare professionals, and the elderly)
	Institutionalized open disclosure	To develop and institutionalize guidelines for open disclosure (e.g., communication and discussion with patients and their families, apologies, and compensation without any penalty regarding disclosure)
System improvement in cause analysis	Development and spread of tools for analyses	To develop and disseminate standardized tools for the cause analysis of safety events
	Establishment of exclusive committee	To establish professional committees that take full responsibility of patient safety issues at a national or institutional level
	Education of healthcare professionals	To provide training to healthcare professionals (e.g., pharmacoepidemiology) to strengthen the individual professionals’ ability to cope with medication errors occurring in their institution
	Integration with IT technology	To develop IT technology such as data mining that detects signals of medication errors using patients medical record
System improvement in counterplan	By regulatory government agency	To prepare government-level countermeasures such as the establishment of alarm systems, providing guidance for the pharmaceutical industry, dissemination of information regarding the safe use of drugs, establishment of a reimbursement system for error reporting (e.g., incentives for good reporting and legal liability for insufficient reporting), and development of a system that help institutions exchange patients information during patient transfer (e.g., medication reconciliation service)
	By pharmaceutical industry	To establish industry-level countermeasures such as making patient brochure, restraint of making similar looking products, and production of pediatric-specific dosage formulation
	By healthcare professionals	To establish professional-level countermeasures such as developing an education program/materials, regular and mandatory education for professional knowledge, introducing courses related to patient safety (e.g., patient safety law and communication skill) in College of Medical, Nursing, and Pharmacy
	By healthcare institutions	To establish industry-level countermeasures such as staff training, regular discussion on errors occurring in the institution, the establishment of computerized physician order entry, improvements to the workflow and work environment, and developing guidelines for providing patient with medication information
	By patients	To establish patient-level countermeasures such as an education program on medication error and participation of patient/caregiver in patient safety committee
System improvement in assessment	Regularization of system assessment	To regularly evaluate the system related to medication error and seek ways to improve the system
	Development and spread of guidelines for assessment	To develop and disseminate the guidelines for assessment (e.g., design, criteria/indices, measurement, and analysis method) to acquire high-quality results

**Table 3 healthcare-10-00512-t003:** Integrated matrix and normalized weights of criteria.

	Cultural	System Reporting	System Cause Analyses	System Counterplan	System Assessment	Geometric Mean	Normalized Weights
Cultural	1.000	1.066	1.763	1.272	2.810	1.464	0.261 *
System reporting	0.938	1.000	1.907	1.070	3.672	1.477	0.263
System cause analyses	0.567	0.524	1.000	0.411	2.946	0.815	0.145
System counterplan	0.786	0.935	2.432	1.000	3.753	1.463	0.261 *
System assessment	0.356	0.272	0.339	0.266	1.000	0.388	0.069
Total						5.607	1.000

* The normalized weight of cultural improvement was larger than that of system improvement in counterplan.

**Table 4 healthcare-10-00512-t004:** Normalized weights of alternatives.

	Normalized Weights	Alternatives	Normalized Weights (within Criterion)	Normalized Weights (Overall)
Cultural	0.261 *	Culture for open disclosure	0.243	0.063
		Participation of laypeople	0.178	0.047
		Change from a blame culture to a safety culture	0.445	0.116
		Facilitating relevant research	0.134	0.035
System reporting	0.263	Establishment of exclusive organization	0.187	0.049
		Building of reporting system	0.391	0.103
		Development and spread of guidelines for reporting	0.160	0.042
		Institutionalized open disclosure	0.262	0.069
System cause analyses	0.145	Development and spread of tools for analyses	0.299	0.044
		Constitution of exclusive committee	0.158	0.023
		Education of healthcare professionals	0.329	0.048
		Integration with IT technology	0.214	0.031
System counterplan	0.261 *	By regulatory government agency	0.118	0.031
		By pharmaceutical industry	0.220	0.057
		By healthcare professionals	0.158	0.041
		By healthcare institutions	0.451	0.118
		By patients	0.052	0.014
System assessment	0.069	Regularization of system assessment	0.558	0.039
		Development and spread of guidelines for assessment	0.442	0.031

* The normalized weight of cultural improvement was larger than that of system improvement in counterplan.

**Table 5 healthcare-10-00512-t005:** Priority of prevention strategies for medication errors.

Rank (Base-Case)	Factors	Weights	Rank (Sensitivity Analyses)	
			Geometric Mean	Arithmetic Mean
**Criteria**				
1	System improvement in reporting	0.263	1	3
2	Cultural improvement	0.261 *	2	1
3	System improvement in counterplan	0.261 *	3	2
4	System improvement in cause analyses	0.145	4	4
5	System improvement in assessment	0.069	5	5
**Alternatives**				
1	Counterplan by healthcare institutions	0.118	1	2
2	Change from a blame culture to a safety culture	0.116	2	1
3	Building of reporting system	0.103	3	4
4	Institutionalized open disclosure	0.069	4	6
5	Culture for open disclosure	0.063	6	3
6	Counterplan by pharmaceutical industry	0.057	5	7
7	Establishment of exclusive organization for reporting	0.049	7	11
8	Education of healthcare professionals for cause analyses	0.048	8	16
9	Participation of laypeople	0.047	12	5
10	Development and spread of tools for cause analyses	0.044	10	12
11	Development and spread of guidelines for reporting	0.042	11	15
12	Counterplan by healthcare professionals	0.041	9	10
13	Regularization of system assessment	0.039	13	8
14	Facilitating relevant research	0.035	15	9
15	Integration of cause analyses and IT technology	0.031 ^†^	17	17
16	Counterplan by regulatory government agency	0.031 ^†^	14	14
17	Development and spread of guidelines for system assessment	0.031 ^†^	16	13
18	Constitution of exclusive committee for cause analyses	0.023	18	18
19	Counterplan by patients	0.014	19	19

* The normalized weight of cultural improvement was larger than that of system improvement in counterplan. ^†^ The normalized weight of integration of cause analyses and IT technology was larger than that of counterplan by regulatory government agency. Additionally, the normalized weight of counterplan by regulatory government agency was larger than that of development and spread of guidelines for system assessment.

## Data Availability

The dataset will be made available upon reasonable request.

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
