# Peer review of "Priorities in the Prevention Strategies for Medication Error Using the Analytical Hierarchy Process Method"

_healthcare, 2022, doi:10.3390/healthcare10030512_

Round 1

Reviewer 1 Report

  1. The authors should provide the eigenvector method to aggregate the pairwise comparisons obtained from experts and to estimate the weights of each criterion and alternative on related work and the current trend research. The authors should have enough background information to provide more acceptable of the proposed method in the paper.
  2. Problem statement that the authors mentioned in the introduction section is quite weak and it cannot convince why we need the partitioned data sets. If the authors identify detailed background, it would make this paper more credible. If the author does so, it would streng then the position of the paper.
  3. The authors need to illustrate that their proposed method can improve interactivity which the authors claimed that this issue is better than previous work that they reviewed. It has the short contents for the performance and results.
  4. The author should mention how to solve the problem of the proposed method for the future research.
  5. Your article need more technology detail in the conclusion part.

Author Response

  1. The authors should provide the eigenvector method to aggregate the pairwise comparisons obtained from experts and to estimate the weights of each criterion and alternative on related work and the current trend research. The authors should have enough background information to provide more acceptable of the proposed method in the paper.

[Authors’ response] We appreciate your comment. For the eigenvector method used in this study, we presented the pairwise comparison matrix and normalized weights in Table 3 and Table S1­–5 in the submitted article. However, we agree with you that there is a need for more information on this approach, thus we added a description on how to estimate the normalized weights in the Methods section.

  • Line 106–107 in the revised manuscript (clean version):

“…We estimated the normalized weights of each criterion and alternative by using geometric mean calculated in every row of the pairwise comparison matrix. …”

For the alternatives on related work and the current trend research, we discussed previous studies using AHP and the strength of MCDA in Line 171–196 in the submitted article. In addition to this, we elaborated our discussion by reviewing the current trend of MCDA and the reason why we used the AHP method in this study to address your comment.

  • Line 226–231 in the revised manuscript (clean version):

“… Therefore, the number of studies using MCDA in healthcare research field has steadily increased between 1990 and 2017, the AHP (41%) was the most widely used method among the MCDA methods, followed by the fuzzy logic (21%) and EVIDEM (12%) [61]. In this study, we used the AHP because it enables structured decision making based on intuitive judgement of experts, reflecting their expertise and experience with patient safety. …”

  1. Problem statement that the authors mentioned in the introduction section is quite weak and it cannot convince why we need the partitioned data sets. If the authors identify detailed background, it would make this paper more credible. If the author does so, it would strengthen the position of the paper.

[Authors’ response] Thank you for your comment. We elaborated the status of medication errors and prevention strategies and the need for using AHP in the Introduction section.

  • Line 41–42 in the revised manuscript (clean version):

“…Medication errors result in considerable economic burden, which was estimated to be £98.5 million per year in England [6].”

  • Line 48–63 in the revised manuscript (clean version):

“… The Food and Drug Administration (FDA) of the United States currently operates the FDA Adverse Event Reporting System (FAERS) as a database that collects drug adverse events and medication errors. The data accumulated in FAERS are analyzed and evaluated by the Center for Drug Evaluation and Research and Center for Biologics Evaluation and Research (CBER) for postmarketing surveillance [9]. The National Health Service of the United Kingdom implemented national campaigns, such as ‘Patient Safety First’ and ‘Sign up to Safety’, and encouraged incident reporting by assuring anonymity and giving an incentive [10]. Moreover, various education programs have been delivered to patients vulnerable to medication errors, including elderly, asthma, and diabetes patients, and demonstrated to be effective for preventing errors [11-13].

Decision makers of patient safety policies have difficulty in setting priorities between various alternatives, considering the relevant evidence and necessity of each alternative, and decision-making becomes more complicated when involving several individuals that have different priorities for the alternatives [14]. To make a decision systematically and transparently, the analytic hierarchy process (AHP) can be utilized. …”

For the comment about partitioned data sets, we do not fully understand what it means. If you can provide more explanation for this, we will look into this point. Assuming that you meant the partitioned data sets as the data coming from two groups (experts on patient safety research and experts on clinical pharmacotherapy), we used the partitioned data sets to get balanced information from each group regarding the patient safety and medication error.

  1. The authors need to illustrate that their proposed method can improve interactivity which the authors claimed that this issue is better than previous work that they reviewed. It has the short contents for the performance and results.

[Authors’ response] We appreciate your comment. We have added explanation on the strengths of AHP approach in the Discussion section.

  • Line 229–235 in the revised manuscript (clean version):

“… In this study, we used the AHP because it enables structured decision making based on intuitive judgement of experts, reflecting their expertise and experience with patient safety. In addition, the AHP has an advantage in its simplicity of application compared to other methodologies that use mathematical models. It uses relative scale (i.e., standardized measurement unit) for pairwise comparisons, enabling efficient data accumulation and further analyses in case of conducting successive research. …”

  1. The author should mention how to solve the problem of the proposed method for the future research.

[Authors’ response] We appreciate your comment. We have added description about the drawback of the AHP approach and suggested future research that can solve this problem in the Discussion section.

  • Line 235–238 in the revised manuscript (clean version):

“… However, as the AHP uses linguistic expression to measure the responder’s decision, it inevitably involves linguistic uncertainty [62]. To overcome this uncertainty, the fuzzy AHP that combines the AHP and fuzzy set theory can be utilized in future research.”

  1. Your article need more technology detail in the conclusion part.

[Authors’ response] Thank you for your comment. We revised the conclusion part to include technology detail of the AHP succinctly.

  • Line 249–251 in the revised manuscript (clean version):

“We have suggested the priority of preventive strategies against medication errors using the AHP method that enables structured decision making based on expertise and experience of patient safety experts. …”

Reviewer 2 Report

Summary: The study looked at the prevention strategies in medication error using the Analytical Hierarchy process. The hierarchy structure consisted of three stages which included the goal of the decision, decision criteria and alternatives. For the decision criteria, system improvement in reporting was the most preferred criterion while the preferred alternative was a counterplan by healthcare institutions.

Introduction: I suggest that the introduction should include be more detailed than what was written. The information was too short and it did not bring out the rationale adequately for the study. Also, situations of medication error in various countries used in the analysis were not highlighted.

Materials and Methods:

Table 1: The table showed that the experts on patient safety research had more work experience and more educated than the experts on clinical pharmacotherapy. I hope this did not affect the judgments. It would have been better to have the same distribution in terms of work experience and education.

Discussion: Lines 157-165- This paragraph was not necessary to be in this section as it was not discussing the results, rather it was talking about the analysis.

The authors did not discuss the results but only wrote about AHP and other methods that had been used before. The results should be discussed extensively by getting literatures that oppose or support the findings of the study.

Author Response

  1. Introduction: I suggest that the introduction should include be more detailed than what was written. The information was too short and it did not bring out the rationale adequately for the study. Also, situations of medication error in various countries used in the analysis were not highlighted.

[Authors’ response] We appreciate your comment. We revised the introduction to include more information and elaborated the situations of medication error in various countries as you suggested.

  • Line 41–42 in the revised manuscript (clean version):

“…Medication errors result in considerable economic burden, which was estimated to be £98.5 million per year in England [6].”

  • Line 48–63 in the revised manuscript (clean version):

“… The Food and Drug Administration (FDA) of the United States currently operates the FDA Adverse Event Reporting System (FAERS) as a database that collects drug adverse events and medication errors. The data accumulated in FAERS are analyzed and evaluated by the Center for Drug Evaluation and Research and Center for Biologics Evaluation and Research (CBER) for postmarketing surveillance [9]. The National Health Service of the United Kingdom implemented national campaigns, such as ‘Patient Safety First’ and ‘Sign up to Safety’, and encouraged incident reporting by assuring anonymity and giving an incentive [10]. Moreover, various education programs have been delivered to patients vulnerable to medication errors, including elderly, asthma, and diabetes patients, and demonstrated to be effective for preventing errors [11-13].

Decision makers of patient safety policies have difficulty in setting priorities between various alternatives, considering the relevant evidence and necessity of each alternative, and decision-making becomes more complicated when involving several individuals that have different priorities for the alternatives [14]. To make a decision systematically and transparently, the analytic hierarchy process (AHP) can be utilized. …”

  1. Table 1: The table showed that the experts on patient safety research had more work experience and more educated than the experts on clinical pharmacotherapy. I hope this did not affect the judgments. It would have been better to have the same distribution in terms of work experience and education.

[Authors’ response] Thank you for your comment. As you pointed out, experts on patient safety research had more work experience and more educated than the experts on clinical pharmacotherapy, because we included key opinion leaders in patient safety research in South Korea. Moreover, we included both types of experts to investigate various aspects of the preventive strategies of medication error and to get balanced information from each group regarding the patient safety and medication error. We agree with you that work experience and education might have influenced the judgements. Therefore, we compared the priorities of criteria between individual experts as below, and it seems that the expertise did not play a role in the judgements.

Experts

Cultural improvement

System –

Reporting

System –

Cause analyses

System –

Counterplan

System –

Assessment

Patient safety research

Expert #1

1

1

3

3

5

Expert #2

5

1

3

1

4

Expert #3

1

2

4

3

5

Expert #4

1

4

3

2

5

Expert #5

2

3

4

1

5

Clinical pharmacotherapy

Expert #6

5

4

3

2

1

Expert #7

1

2

4

3

5

Expert #8

5

2

2

1

4

Expert #9

1

2

2

4

5

Expert #10

2

1

3

4

5

  1. Discussion: Lines 157-165- This paragraph was not necessary to be in this section as it was not discussing the results, rather it was talking about the analysis.

[Authors’ response] Thank you for pointing this out. We moved the paragraph to the Method and the first paragraph of the Discussion where appropriate.

  • Line 123–126 in the revised manuscript (clean version):

“…The approach in the base-case analysis is primarily used when information or previous research on the topic is rare, and the other approaches are used to place an emphasis on the expertise of individual responders involved in decision making [54]. …”

  • Line 181–182 in the revised manuscript (clean version):

“… The results of the sensitivity analyses indicated that the priorities of decision criteria and alternatives were generally robust to aggregating approaches. …”

  1. The authors did not discuss the results but only wrote about AHP and other methods that had been used before. The results should be discussed extensively by getting literatures that oppose or support the findings of the study.

[Authors’ response] We appreciate your comment. We have added a paragraph to review previous studies and compare with our results in the Discussion section as you suggested.

  • Line 183–200 in the revised manuscript (clean version):

“To our knowledge, this is the first study to provide evidence for priority setting in prevention strategies against medication errors using the AHP method. Although several studies have investigated strategies for the prevention of medication errors, most of them discussed the strategies in a descriptive way based on the literature review, and a few of them evaluated the priority of strategies using survey data. Matti et al. conducted a survey of staffs at neonatal intensive care units in Australia and New Zealand to understand the prevention strategies utilized in clinical practice [55]. The survey found that smart infusion pumps, ward-based pharmacists, and administration calculation test for nursing staffs were the most frequently used strategies to prevent medication errors. Fortescue et al. examined the types of medication errors and strategies that might have prevented the potentially harmful errors in 1,020 pediatric inpatients [56]. The study reported that the most effective interventions to prevent potentially harmful errors were ward-based pharmacists, improving communication between healthcare providers, and computerized physician order entry combined with clinical decision support system. Contrary to previous studies, our study assessed the priority of prevention strategies at a national level rather than an institutional level. However, our study demonstrated that counterplan by healthcare institutions was the most important alternative, highlighting the need for developing evidence-based strategies for healthcare institutions.”

Round 2

Reviewer 1 Report

When I have checked the paper, the author is most satisfied the paper for requirements. However, it needs to check english grammar and journal structure. English grammar and structure were improving than previous version.